

# Ferromagnetism of LaCoO$_3$ films

**Andrii Sotnikov[1,2⋆], Kyo-Hoon Ahn[1,3] and Jan Kuneš[1,4]**

**1** Institute of Solid State Physics, TU Wien, Wiedner Hauptstraße 8, 1020 Vienna, Austria
**2** Akhiezer Institute for Theoretical Physics,
NSC KIPT, Akademichna 1, 61108 Kharkiv, Ukraine
**3** Division of Display and Semiconductor Physics, Korea University, Sejong 30019, Korea
**4** Institute of Physics, Czech Academy of Sciences, Na Slovance 2, 182 21 Praha 8, Czechia

⋆ sotnikov@ifp.tuwien.ac.at

## Abstract

We study ferromagnetic ordering and microscopic inhomogeneity in tensile strained LaCoO$_3$ using numerical simulations. We argue that both phenomena originate from effective superexchange interactions between atoms in the high-spin (HS) state mediated by the intermediate-spin excitations. We derive a model of the HS excitation as a bare atomic state dressed by electron and electron-hole fluctuations on the neighbor atoms. We construct a series of approximations to account for electron correlation effects responsible for HS fluctuations and magnetic exchange. The obtained amplitudes and directional dependence of magnetic couplings between the "dressed" HS states show a qualitative agreement with experimental observations and provide a new physical picture of LaCoO$_3$ films.

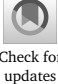
## 1   Introduction

$LaCoO_3$ is an non-magnetic insulator, which develops a Curie-Weiss magnetic response and eventually transforms to metal with increasing temperature [1–4]. This peculiar behavior arising from proximity of several atomic multiplets has attracted attention for decades and the debate is not over yet. Strained films of $LaCoO_3$ were shown to develop a ferromagnetic (FM) order [5–9] while remaining insulating [10]. Small value of saturated magnetization suggests that the FM state is not uniform on the atomic scale. Fujioka *et al.* [11] observed a commensurate structure with the propagation vector (1/4, -1/4, 1/4). A superstructure with magnetic atoms separated by three non-magnetic ones was observed with transmission electron microscopy [12].

The discussion of bulk $LaCoO_3$ revolves around the nature of the lowest atomic excited state of $Co^{3+}$ with the alternatives being the high-spin (HS) or the intermediate-spin (IS) states [13–17]. The spectroscopic data [18,19] as well as crystal-field considerations [20] clearly favor the HS state as the lowest excitation. Two of us have recently proposed a new paradigm for the bulk compound [21] arguing that the $Co^{3+}$ IS states in a crystal environment of $LaCoO_3$ cannot be viewed as atom-bound objects, but must be treated as mobile bosons (excitons) reflecting the mobility of the constituting $e_g$ electron and $t_{2g}$ hole. Recent resonant inelastic x-ray scattering (RIXS) experiments [22,23] reveal sizable dispersion of IS excitons supporting this picture. As a result, the IS states cannot be neglected despite their single-ion energy being several 100 meV above the low-spin (LS) ground state.

A peculiar feature of $LaCoO_3$ is the short-range ferromagnetic correlation in the middle-temperature range [24,25], which becomes more pronounced in thin films leading to a ferromagnetic phase below 94 K [11]. Should the magnetism of $LaCoO_3$ originate from the HS state of $Co^{3+}$ ions only, an anti-ferromagnetic (AFM) coupling between these is expected [26]. There were several attempts to explain emerging FM long-range order in insulating $LaCoO_3$ under tensile strain [25,27–30]. The LDA+U density functional calculations using Hartree-Fock-like treatment of intra-atomic $d$-$d$ interaction [28,29] argue in favor of the superexchange-based mechanism for FM ordering. However, a more detailed analysis within the same framework allowing for AFM arrangement of HS shows that the LDA+U treatment can not be considered as conclusive due to diverse results, which strongly depend on the employed interaction parameters (see Ref. [29] and Appendix A for more details).

In this paper, we build a model of $LaCoO_3$ in the strained tetragonal structure with the aim to explain its ferromagnetism and atomic-scale inhomogeneity. The key microscopic observations are: i) The HS excitations can be viewed as bound pairs of mobile IS excitons. ii) The IS-IS bonding is strong enough to make HS bi-exciton a compact and almost immobile object. iii) At the same time, the IS-IS bonding is weak enough to allow existence of a cloud of virtual IS excitations on the Co sites neighboring the site in the HS state. iv) The interactions of overlapping clouds surrounding different HS sites are decisive for the HS-HS interactions beyond the nearest neighbors. These interactions are strongly anisotropic reflecting the shapes of the relevant Co $d$ orbitals. These microscopic features are key to understanding the properties of $LaCoO_3$ films.

Despite $LaCoO_3$ being an insulator with quenched charge fluctuations, the number of lo-

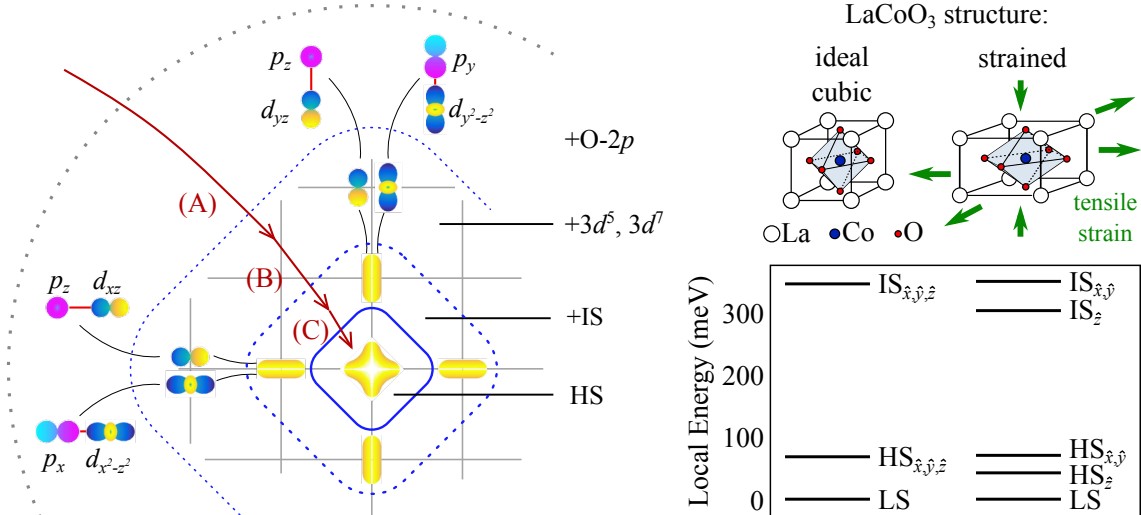

Figure 1: (Left) Schematic representation of the main steps performed in the analysis with a gradual elimination of the high-energy states. (Right) Schematic comparison of the lowest local energies obtained after the step (B) for the depicted cubic (bulk) and strained (film) LaCoO$_3$ structures.

cal low-energy degrees of freedom involving LS, IS, and HS states is prohibitively large for a direct numerical treatment. Therefore, we construct a series of approximations. Starting from electronic description, we integrate out the charge fluctuations to arrive at an effective model in the Hilbert space spanned by the LS, IS, and HS states [21, 22]. We take advantage of the tetragonal crystal field, which lifts the orbital degeneracy of the HS and IS states and thus simplifies the description of the strained LaCoO$_3$ films in comparison to the bulk material. This model describes a gas of IS excitonic particles with strongly-directional mobility depending on their orbital character. Strong local attraction between IS excitons of different orbital character renders their immobile bound pairs (HS states) to be the lowest excitations of the system. In the next step, we integrate out the IS excitons and formulate a model in terms of HS particles. Dressed in that way HS bi-excitons possess strongly-directional inter-site interaction varying between repulsion along the crystallographic axes and in-plane next-nearest-neighbor attraction in the diagonal direction. As a result, spatially separated chains of HS sites provide an energetically stable building block. We show that dynamics of such chain can be described by the Ising-like model with FM coupling, which provides the lowest level of simplification we arrive at. We further study the arrangement of such chains in a three-dimensional (3D) system.

## 2 Theoretical approach

Our approach consists of three successive steps (see also Fig. 1):

(A) Density-functional theory (DFT) calculation is performed for the strained LaCoO$_3$ structure [31]. A tight-binding ($d$-only) model in Wannier basis spanning the Co $d$-like bands is constructed [32, 33] and augmented with the local Coulomb interaction within the Co $3d$ shell, which amounts to projecting out the O-$2p$ orbitals.

(B) The 5-orbital Hubbard model is reduced to an effective low-energy model spanning the LS, HS and IS states by employing the Schrieffer-Wolff transformation [34]. This amounts to integrating out the local charge fluctuations from $d^6$ to $d^5$ and $d^7$ configura-

tions. Furthermore, we use the tetragonal symmetry to restrict our model to the lowest HS-orbital singlet and two orbital flavors of the IS multiplet that are coupled. The model at this level describes a gas of hard-core IS excitons, which propagate on the lattice. The IS excitons with different orbital flavors interact via strong local attractive interaction, which gives rise to local bound pairs, bi-excitons.

(C) Analysing the behavior of the model (B), we come to the conclusion that for realistic strength of the attractive IS-IS interaction, the local bi-excitons are stable objects consisting of atomic HS state dressed with IS-IS fluctuations to the nearest-neighbor (nn) sites. The last layer of approximation consists in integrating out the IS states. This way we arrive at a model formulated in terms of HS particles only. These turn out to be immobile in the $xy$ plane and interacting via strongly directional interactions.

In the following, we provide technical details about execution of the above program and point out the special features of the models (A)-(C), which justify our treatment. The numerical results and their discussion is left for the next section. The present theory neglects the structural relaxation in the microscopically inhomogeneous state, which may be important to capture its stability [35–38].

Step (A) follows a standard approach of constructing the multi-orbital Hubbard model used for correlated materials [39]. DFT computational parameters are summarized in Appendix A. To incorporate the correlation effects originating from electron interactions in the $Co^{3+}$ $d^6$ shell of $LaCoO_3$, we introduce the 5-orbital Hubbard Hamiltonian

$$\mathcal{H}_A = \sum_i \mathcal{H}_{\text{at}}^{(i)} + \mathcal{H}_t,$$
$$\mathcal{H}_{\text{at}}^{(i)} = \sum_\kappa \varepsilon_\kappa c_{i\kappa}^\dagger c_{i\kappa} + \mathcal{H}_{\text{SOC}} + \sum_{\kappa\lambda\mu\nu} U_{\kappa\lambda\mu\nu} c_{i\kappa}^\dagger c_{i\lambda}^\dagger c_{i\nu} c_{i\mu}. \tag{1}$$

The local part $\mathcal{H}_{\text{at}}^{(i)}$ consists of the tetragonal crystal field (diagonal in cubic harmonics basis), spin-orbit coupling (SOC), and the Coulomb interaction, while $\mathcal{H}_t$ describes the inter-atomic hopping. The coupling constants are given in Appendix B, $c_{i\kappa}^\dagger$ ($c_{i\kappa}$) are the fermionic creation (annihilation) operators, $i$ labels the cobalt ion sites, while $\kappa, \lambda, \mu, \nu$ are the combined orbital and spin state indices.

In following, we proceed without spin-orbit interaction $\mathcal{H}_{\text{SOC}}$, which we treat perturbatively as described later. This way we can use higher symmetry of the Hamilatonian and apply corrections at the end instead of neglecting small terms at various points of steps (B) and (C). We diagonalize the local Hamiltonian $\mathcal{H}_{\text{at}}^{(i)}$ and define the low-energy subspace $\mathcal{L}$ to be the space spanned by 25 lowest $d^6$ states: the spin-singlet $S = 0$ LS state, the spin-triplet $S = 1$ IS states with the $t_{1g}$ orbital symmetry ($IS_{\hat{x}} \equiv (y^2 - z^2) \otimes yz$, $IS_{\hat{y}} \equiv (z^2 - x^2) \otimes zx$, $IS_{\hat{z}} \equiv (x^2 - y^2) \otimes xy$), and the spin $S = 2$ HS states with the $t_{2g}$ orbital symmetry ($HS_{\hat{x}} \equiv yz$, $HS_{\hat{y}} \equiv zx$, $HS_{\hat{z}} \equiv xy$). The strong-coupling expansion consists in the Schrieffer-Wolff transformation to the second order in the hopping $\mathcal{H}_t$ [40] (we consider the nn processes only). The resulting effective Hamiltonian can be written in the form

$$\mathcal{H}_B = \sum_{\mathbf{i},\mathbf{e}} \sum_{\alpha,\beta,\gamma,\delta \in \mathcal{L}} \varepsilon_{\alpha\beta\gamma\delta}^{(\mathbf{e})} T_{\mathbf{i}}^{\alpha\beta} T_{\mathbf{i}+\mathbf{e}}^{\gamma\delta}, \tag{2}$$

where the operator $T_{\mathbf{i}}^{\alpha\beta}$ describes a transition between local states $|\beta\rangle$ and $|\alpha\rangle$ on the site $\mathbf{i}$, while $\mathbf{e}$ labels the nn bonds.

The number of local degrees of freedom in the above model is still too high to allow for a numerical treatment of even a small cluster. We notice several symmetries that are weakly broken by SOC: (i) Viewing the HS state as a pair of IS excitons, the number of IS excitons of

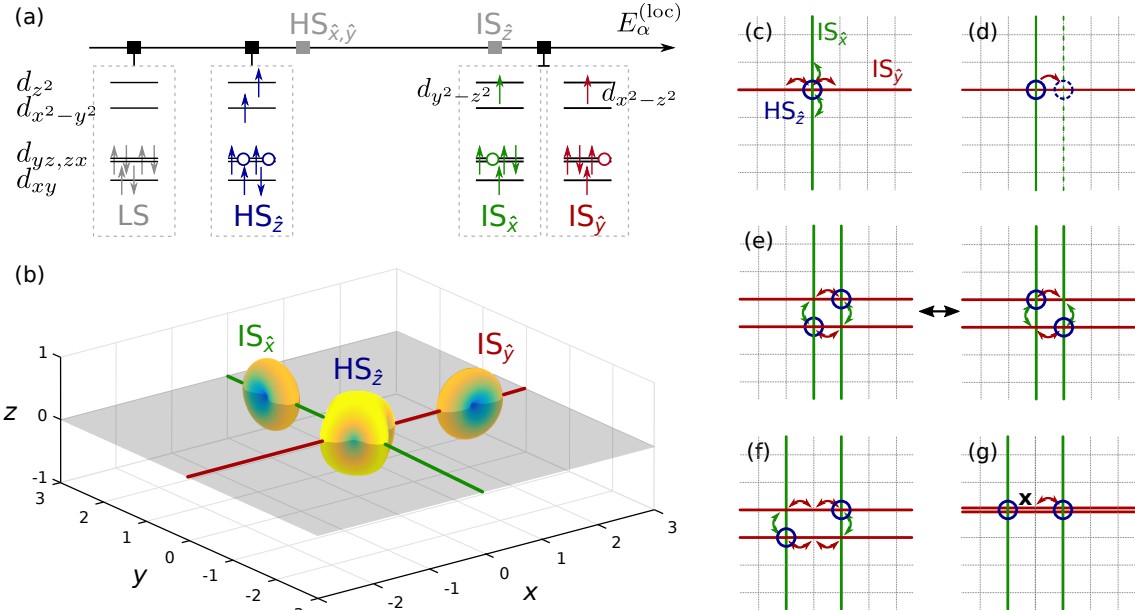

Figure 2: (a) Spin configurations of the relevant atomic $d^6$ states with tentative positions of their local energies $E_\alpha^{(\text{loc})}$ in the strained LaCoO$_3$. (b) Isosurfaces of the electron charge density from active (singly-occupied) orbitals of excitons in the lattice. (c)-(g) Schematic representation of spatial configurations studied within the ED approach in the $xy$ plane. The mobile IS excitations are represented by lines and HS bi-excitons by open circles at their intersections.

a given orbital character is conserved. This is a simple consequence of the fact that a $t_{2g}$-hole cannot change its character (nn hopping is only possible between the same $t_{2g}$ orbitals). This will allow us to work in subspaces indexed by the total number of IS excitons. (ii) The hopping amplitude of IS$_{\hat{\alpha}}$ along the $\alpha$-axis is very small. We will assume that this hopping vanishes completely and thus IS$_{\hat{\alpha}}$ are confined to planes perpendicular to the $\alpha$-axis. (iii) HS state can be viewed as a bi-exciton, e.g, HS$_{\hat{z}} \equiv$ IS$_{\hat{x}} \otimes$ IS$_{\hat{y}}$. Non-orthogonality of the $e_g$ orbitals forming the two IS excitons is accounted for by a correlated hopping terms in the effective Hamiltonian. (iv) The tetragonal crystal field splits HS$_{\hat{z}}$ from the set of HS states to be the lowest state. In the following, we will consider the subspace containing only HS$_{\hat{z}}$ and the states coupled to it, i.e., IS$_{\hat{x}}$ and IS$_{\hat{y}}$ (see also Fig. 2). This choice is justified *a posteriori* by noting that for realistic parameters, the lowest excited states are HS-like. (v) The Hamiltoninan (2) without SOC has the SU(2) spin symmetry, which will allow us to further restrict the Hilbert space in numerical calculations.

The coupling constants of the model (2) consist of nn coupling and on-site energies. The nn coupling is determined by the products of hopping elements in $\mathcal{H}_t$ divided by denominator of the order of the interaction strength $F_0$. Changing the parameters of local interaction leads to more or less uniform scaling of all nn couplings, which does not change the studied physics significantly. Therefore, we treat the nn couplings as fixed, see Appendix B for details. The on-site energies of the IS and HS excitations, which are obtained by the strong-coupling expansion and measured relative to the LS background, are less reliable, since they are calculated from differences of large numbers. In particular, the lowest excitation energy is experimentally known to be rather small ($< 20$ meV). For this reason we fix the on-site energy of the IS$_{\hat{x},\hat{y}}$ exciton to $E_{\text{IS}} = 330$ meV, which matches well the RIXS data for the bulk LaCoO$_3$, and treat the on-site IS-IS attraction $U$, $E_{\text{HS}} = 2E_{\text{IS}} - U$, as adjustable parameter. Since all our calculations are performed for a fixed number of IS excitons, the results do not depend on $E_{\text{IS}}$. This way

we can analyze relative stability of various states, while the absolute excitation energy above the singlet ground state is not studied.

The step between models (B) and (C) is executed by performing series of exact diagonalization (ED) calculations for small clusters. The computational effort is heavily reduced by the confinement of IS excitons into their respective planes (or even lines in calculations for a single $xy$ plane). We start by investigating the structure of a single HS excitation. To this end, we perform calculations in the subspace with one of each $IS_{\hat{x}}$ and $IS_{\hat{y}}$ exciton. We find that for realistic values of $U$, the ground state in the 2D case is a HS state located on the intersection of the "life-lines" of the two IS excitons, see Fig. 2(b) and (c), with some admixture of IS-IS configurations. In the 3D case, the "life-lines" turn into "life-planes" and the HS state with the cloud of IS-IS fluctuations can propagate along a vertical line parallel to the $z$-axis in Fig. 2(b).

Next, we study the effect of SOC in the single HS set-up. We find that it can be approximated by replacing the spin $S = 2$ and the isotropic exchange $\mathbf{S}_i \cdot \mathbf{S}_j$ with a pseudospin $J = 1$ anisotropic $J$-$J$ exchange with stronger in-plane component.

Finally, we study clusters with pairs of HS bi-excitons, i.e., the subspaces containing two of each $IS_{\hat{x}}$ and $IS_{\hat{y}}$ excitons. We determine that the lowest states are linear combinations of the dressed HS bi-excitons and use the respective eigenenergies to determine parameters of the model (C) formulated in terms of HS states only. In addition to the bare nearest-neighbor HS-HS repulsion present in the model (B), we obtain longer-range interactions, which arise from the interaction of the IS-IS clouds surrounding the HS cores.

# 3 Results and discussion

## 3.1 HS-IS model

Upon the Schrieffer-Wolff transformation of the model (1) and restricting to the lowest-energy sector containing $HS_{\hat{z}}$, the Hamiltonian (2) acquires the form

$$
\begin{aligned}
\mathcal{H}_B = {} & E_{IS} \sum_{\mathbf{i},\Lambda,s} b^{\dagger}_{\mathbf{i},\Lambda s} b_{\mathbf{i},\Lambda s} + (2E_{IS} - U) \sum_{\mathbf{i},s} h^{\dagger}_{\mathbf{i},s} h_{\mathbf{i},s} + \sum_{\lambda,\Lambda} t_{\lambda\Lambda} \sum_{\mathbf{i},s} b^{\dagger}_{\mathbf{i}\pm e_{\lambda},\Lambda s} L_{\mathbf{i}\pm e_{\lambda}} L^{\dagger}_{\mathbf{i}} b_{\mathbf{i},\Lambda s} \\
& + \sum_{\lambda,\Lambda} t^{c}_{\lambda\Lambda} \sum_{\mathbf{i},s,s'} B_{ss'} h^{\dagger}_{\mathbf{i}\pm e_{\lambda},s+s'} b_{\mathbf{i}\pm e_{\lambda},\bar{\Lambda}s'} L^{\dagger}_{\mathbf{i}} b_{\mathbf{i},\Lambda s} + \text{H.c.} \\
& + \sum_{\mathbf{i},\lambda,\Lambda} \sum_{k=0}^{2} \tilde{t}^{(k)}_{\lambda\Lambda} \sum_{q=-k}^{k} (-1)^q \sum_{s,s'} K^{(k)}_{ss+q} C^{(k)}_{s's'-q} h^{\dagger}_{\mathbf{i}\pm e_{\lambda},s} b_{\mathbf{i}\pm e_{\lambda},\bar{\Lambda}s'-q} b^{\dagger}_{\mathbf{i},\bar{\Lambda}s'} h_{\mathbf{i},s+q} \\
& + \sum_{\mathbf{i},\lambda,\Lambda,\Lambda'} \sum_{k=0}^{2} J^{(k)}_{1\lambda\Lambda\Lambda'} \sum_{q=-k}^{k} (-1)^q \sum_{ss'} C^{(k)}_{ss+q} C^{(k)}_{s's'-q} b^{\dagger}_{\mathbf{i}+e_{\lambda},\Lambda s} b_{\mathbf{i}+e_{\lambda},\Lambda s+q} b^{\dagger}_{\mathbf{i},\Lambda's'} b_{\mathbf{i},\Lambda's'-q} \\
& + \sum_{\mathbf{i},\lambda,\Lambda} \sum_{k=0}^{2} J^{(k)}_{2\lambda\Lambda} \sum_{q=-k}^{k} (-1)^q \sum_{ss'} K^{(k)}_{ss+q} C^{(k)}_{s's'-q} h^{\dagger}_{\mathbf{i}\pm e_{\lambda},s} h_{\mathbf{i}\pm e_{\lambda},s+q} b^{\dagger}_{\mathbf{i},\Lambda s'} b_{\mathbf{i},\Lambda s'-q} \\
& + \sum_{\mathbf{i},\lambda} \sum_{k=0}^{4} J^{(k)}_{3\lambda} \sum_{q=-k}^{k} (-1)^q \sum_{ss'} K^{(k)}_{ss+q} K^{(k)}_{s's'-q} h^{\dagger}_{\mathbf{i}+e_{\lambda},s} h_{\mathbf{i}+e_{\lambda},s+q} h^{\dagger}_{\mathbf{i},s'} h_{\mathbf{i},s'-q} \quad ,
\end{aligned}
\tag{3}
$$

where $L^{\dagger}_{\mathbf{i}}$, $b^{\dagger}_{\mathbf{i},\Lambda s}$ and $h^{\dagger}_{\mathbf{i},s}$ are creation operators of Schwinger bosons representing LS, $IS_{\Lambda=\hat{x},\hat{y}}$ and $HS_{\hat{z}}$ states, respectively, with the spin projection $s$ and orbital flavor $\Lambda$ on the lattice site $\mathbf{i}$. The physical states fulfil the hard-core constraint $L^{\dagger}_{\mathbf{i}} L_{\mathbf{i}} + \sum_{\Lambda,s} b^{\dagger}_{\mathbf{i},\Lambda s} b_{\mathbf{i},\Lambda s} + \sum_{s} h^{\dagger}_{\mathbf{i},s} h_{\mathbf{i},s} = 1$. The

Table 1: Coupling constants in eV units of the effective model (3). Following symmetries are obeyed: $J^{(k)}_{1\lambda\Lambda\Lambda'} = J^{(k)}_{1\lambda\Lambda'\Lambda}$, $J^{(k)}_{1\bar{\lambda}\bar{\Lambda}\bar{\Lambda}'} = J^{(k)}_{1\lambda\Lambda\Lambda'}$, $J^{(k)}_{2\bar{\lambda}\bar{\Lambda}} = J^{(k)}_{2\lambda\Lambda}$ ($\bar{z} = z, \bar{x} = y, \bar{y} = x$), $t_{\Lambda\bar{\Lambda}} = t_{\bar{\Lambda}\Lambda}$, $t^c_{\Lambda\bar{\Lambda}} = t^c_{\bar{\Lambda}\Lambda}$, $\tilde{t}^{(k)}_{\Lambda\bar{\Lambda}} = \tilde{t}^{(k)}_{\bar{\Lambda}\Lambda}$, and $t_{\Lambda\Lambda} = t^c_{\Lambda\Lambda} = \tilde{t}^{(k)}_{\Lambda\Lambda} = 0$.

| $t_{x\hat{y}}$ | $t_{z\Lambda}$ | $t^c_{x\hat{y}}$ | $t^c_{z\Lambda}$ | $\tilde{t}^{(0)}_{x\hat{y}}$ | $\tilde{t}^{(1,2)}_{x\hat{y}}$ | $\tilde{t}^{(0)}_{z\Lambda}$ | $\tilde{t}^{(1,2)}_{z\Lambda}$ |
|---|---|---|---|---|---|---|---|
| 0.066 | 0.060 | 0.074 | 0.051 | 0.025 | 0.019 | 0.013 | 0.010 |

| $k$ | $J^{(k)}_{1x\hat{x}\hat{x}}$ | $J^{(k)}_{1x\hat{x}\hat{y}}$ | $J^{(k)}_{1x\hat{y}\hat{y}}$ | $J^{(k)}_{1z\hat{x}\hat{x}}$ | $J^{(k)}_{1z\hat{x}\hat{y}}$ | $J^{(k)}_{2x\hat{x}}$ | $J^{(k)}_{2x\hat{y}}$ | $J^{(k)}_{2z\hat{x}}$ | $J^{(k)}_{3x}$ | $J^{(k)}_{3z}$ |
|---|---|---|---|---|---|---|---|---|---|---|
| 0 | -0.001 | -0.017 | 0.175 | 0.074 | 0.054 | -0.008 | 0.184 | 0.116 | 0.185 | 0.179 |
| 1 | -0.001 | -0.033 | 0.091 | 0.036 | 0.018 | -0.012 | 0.045 | 0.031 | 0.023 | 0.027 |

coefficients

$$
\begin{aligned}
B_{ss'} &= \left( \begin{array}{cccc} 1 & s & 1 & s' \end{array} \middle| \begin{array}{cc} 2 & s+s' \end{array} \right), \\
C^{(k)}_{ss'} &= f^{(1,k)}\left( \begin{array}{cccc} 1 & s & k & s'-s \end{array} \middle| \begin{array}{cc} 1 & s' \end{array} \right), \\
K^{(k)}_{ss'} &= f^{(2,k)}\left( \begin{array}{cccc} 2 & s & k & s'-s \end{array} \middle| \begin{array}{cc} 2 & s' \end{array} \right),
\end{aligned}
$$

are normalization factors $f^{(S,2)} = \sqrt{S(S+1)(2S-1)(2S+3)}/3$, $f^{(S,1)} = \sqrt{S(S+1)}$, and $f^{(S,0)} = 1$ times the Clebsch-Gordan coefficients, which ensure the $SO(3)$ spin symmetry.

The summation bounds of the spin index are $s = -1, 0, 1$ and $s = -2, \ldots, 2$ for IS and HS states, respectively. It is also assumed that any expression with $s$ out of the given bounds has zero prefactor. The index $\lambda = x, y, z$ labels directions on the cubic lattice. In the following, we use the number operators $n_{\mathbf{i},a} = \sum_s a^\dagger_{\mathbf{i},s} a_{\mathbf{i},s}$ and $n_a = \sum_{\mathbf{i}} n_{\mathbf{i},a}$. The coupling constants are given in Table 1.

The model (3) has several specific features mentioned above: i) The total numbers $n_{b_{\hat{x}}} + n_h$ and $n_{b_{\hat{y}}} + n_h$ are conserved. ii) The HS $h$-particles do not hop directly. iii) The IS $b_{\hat{x}}$- and $b_{\hat{y}}$-particles can hop only within the $yz$ and $xz$ planes, respectively.

While the Hamiltonian (3) looks complicated, its basis structure is actually simple. It describes two types of excitonic particles $b^\dagger_{\mathbf{i},\Lambda s}$ ($\Lambda = \hat{x}, \hat{y}$), which can move on the lattice in mutually orthogonal planes, interact via strong attractive ferromagnetic on-site interaction $U$ and spin-dependent nn interaction. Since the HS state is not exactly a product of two IS states, the hopping and nn interaction amplitudes to some extent vary depending on the initial state. Therefore, we have $t$ for the $|\text{LS}, \text{IS}\rangle \longleftrightarrow |\text{IS}, \text{LS}\rangle$ process, $t^c$ for $|\text{HS}, \text{LS}\rangle \longleftrightarrow |\text{IS}, \text{IS}\rangle$ process and $\tilde{t}$ for $|\text{HS}, \text{IS}\rangle \longleftrightarrow |\text{IS}, \text{HS}\rangle$ process. Similarly, the nn terms with $J_1$, $J_2$, and $J_3$ correspond to IS-IS, HS-IS, and HS-HS interactions, respectively. The superscript index of $J^{(k)}$ refers to spin-multipole order of the interaction with $k = 0$ corresponding to the $n_a n_b$ density-density interaction, $k = 1$ to $\mathbf{S} \cdot \mathbf{S}$ dipole-dipole interaction. Symmetry allows also quadrupole $k = 2$ interactions, which cannot appear in the second-order processes and thus are absent in our treatment.

Finally, we briefly discuss the physics underlying the coupling constants in Table 1. The planar nature of the exciton hopping refers to the planar nature of both the hole and electron part of the exciton. For example, the $\text{IS}_{\hat{x}}$ exciton consists of $y^2 - z^2$ and $yz$ electron and hole part, respectively, both of which have small hopping amplitude along the $x$ axis (see also Fig. 2). Positive sign of the hopping amplitude corresponds to the excitonic-band maximum at $\Gamma$ point, in agreement with the experiment [22]. To understand the origin of nn interactions, it is necessary to note that on the LS background the IS and HS (bi-)excitons reduce their energy by virtual hopping of electrons and holes to the nn sites. For the HS bi-excitons these processes happen in all directions, while for IS excitons in the respective "life-planes" only. Placing (bi-)excitons of the same orbital character on the nn sites blocks some of these processes, thus

causes an effective repulsion, e.g., $J^{(0)}_{1x\hat{y}\hat{y}}$, $J^{(0)}_{2x\hat{y}}$, or $J^{(0)}_{3x}$. Should the spins of these nn (bi-)excitons be anti-parallel, the Pauli blocking is less effective, which adds an antiferromagnetic dipole-dipole component $J^{(1)}_{1x\hat{y}\hat{y}}$, $J^{(1)}_{2x\hat{y}}$, and $J^{(1)}_{3x}$. The interaction on the bonds, along which one or both of the excitons have vanishing hopping amplitudes, is weak, $J^{(k)}_{1x\hat{x}\hat{x}}$ being example of the latter.

## 3.2 Structure of a single HS state

**Quantum fluctuations.** We begin with numerical analysis of the structure of a single $HS_{\hat{z}}$ excitation. To this end, we investigate the subspace of the total spin $S = 2$ containing one of each $IS_{\hat{x}}$ and $IS_{\hat{y}}$ excitons. Since the IS excitons are constrained to mutually perpendicular planes, the 3D problem is reduced to two intersecting 2D planes, each containing one IS excitation. Similarly, the 2D problem of a single $xy$ plane, depicted in Fig. 2(b), reduces to two intersecting 1D lines. We use exact diagonalization for finite systems with linear dimension $L = 8$ and periodic boundary conditions. The physics is governed by the ratio of the attractive interaction to hopping $U/t$. In Figure 3, we show the energies as functions of $U/t$ (equivalently, $E_{HS}$), as well as the density distributions in the ground state of the 2D and 3D model. For weak $U$, the ground state of the 3D model consists of delocalized (plane-wave) IS excitons living in their respective life-planes. Between $U/t$ of 4 and 6, a bound $HS_{\hat{z}}$ bi-exciton is formed that can propagate along the intersection of the two planes forming thus a narrow band.

What is a realistic magnitude of the on-site IS-IS interaction $U$ in $LaCoO_3$ and how accurately should it be determined? While the excitation gap of $LaCoO_3$, because of its smallness, is very sensitive to the size of $U$, the wave functions of the excitations and their mutual interactions are much less sensitive. Setting $U$ to approximate the experimental energy $\tilde{E}_{HS}$ of HS excitation thus provides a solid starting point for analysis of interactions between excitons. The corresponding value of $\tilde{E}_{HS}$ in bulk $LaCoO_3$ is about 10-20 meV [18, 19]. The gap for elementary excitations in the strained $LaCoO_3$ films is smaller or even negative [1]. In both cases, $|\tilde{E}_{HS}| \ll E_{IS}$, with the experimental value of $E_{IS}$ between 300–350 meV [22]. We can therefore safely assume that $\tilde{E}_{HS} \approx 0$ in our considerations. With some provision for the uncertainty of $E_{IS}$, $\tilde{E}_{HS}$ calculated at $E_{IS} = 330$ meV puts the realistic $U/t$ to the 8–10 interval, see Fig. 3.

In the determined $U/t$ range, the ground state of the 2D and 3D problems is a HS bi-exciton with $|HS, LS\rangle \leftrightarrow |IS, IS\rangle$ quantum fluctuations confined to the nn bonds. The ground state energy $\tilde{E}_{HS} = 2E_{IS} - U - \Delta_Q$ is dominated by the bare HS contribution $2E_{IS} - U$ with a correction $\Delta_Q$ of about 40 meV and 80 meV due to quantum fluctuations in the 2D and 3D cases, respectively. The difference comes from effectively doubling the number of accessible neighbors by going from 2D to 3D. While in the 2D case the HS bi-exciton is confined to the intersection of 1D life-lines of the IS excitons, in the 3D case it can hop along the line of intersection of the corresponding life-planes with a moderate amplitude of +10 meV. Inclusion of SOC lowers $\tilde{E}_{HS}$ by approximately 20 meV, an amount that does not affect the presented estimate of $U$.

The main result so far is the observation that in an insulator the HS bi-exciton is stable and can be viewed as an atomic HS state dressed with IS-IS fluctuations confined to the nn sites. With this picture in mind we will discuss the effect of SOC in the following.

**Spin-orbit coupling.** The SOC introduces an on-site mixing between LS and IS, IS and HS states as well as mixing within the HS and IS multiplets. The former causes violation of the conservation law $2n_{HS} + n_{IS} = \text{const}$. Thanks to the large separation between $E_{IS}$ on one hand and $E_{HS} \approx E_{LS}$ on the other, the violation is weak and will be neglected in the following. The

---

[1]The population of HS or IS states in their ferromagnetic ground state reflects either negative energy of these elementary excitations or an attractive interaction between them.

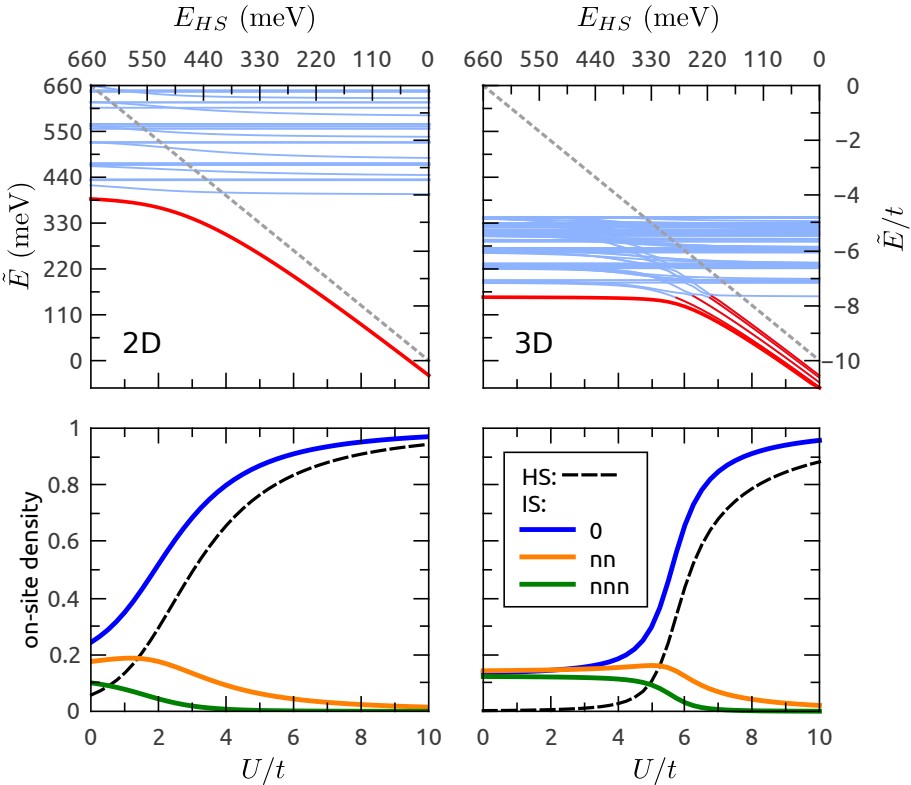

Figure 3: Top row: Energies of the lowest eigenstates in the subspace with one $HS_{\hat{z}}$ state as a function of the IS-IS attraction $U$. Dashed line marks the bare energy $E_{HS}$ assuming $E_{IS} = 330$ meV. The calculations were performed in the geometry described in the text with the linear size $L = 8$. Bottom row: The $IS_{\hat{x}}$ ($IS_{\hat{y}}$) occupancy in the ground state as a function of $U/t$. Site 0 is the crossing point (the HS site). The densities on the nn and next-nearest-neighbour (nnn) sites are calculated along the $x$ ($y$) axis. In the 3D case, the density values integrated over the $z$ direction are shown.

latter makes the hopping of IS excitons and thus HS bi-excitons weakly three-dimensional, however, the effect is much weaker compared to the inter-site interaction, derived in the next section, and is also neglected.

Thanks to the sizeable tetragonal crystal field, the triplet states retain a dominant $HS_{\hat{z}}$ character. The leading effect of SOC is splitting of the $HS_{\hat{z}}$ quintuplet into a ground state quasi-triplet and an excited doublet with energy separation of about 20 meV, see Fig. 4(a). Projection on the quasi-triplet state, while neglecting the admixture of $HS_{\hat{x}}$ and $HS_{\hat{y}}$, approximately amounts to elimination of the $S_z = \pm 2$ states of the $HS_{\hat{z}}$ quintuplet. We introduce a pseudospin $J = 1$ to describe the ground state triplet, which leads to a substitution $S_z \rightarrow J_z$, $S_{x,y} \rightarrow \sqrt{3} J_{x,y}$. The rotationally invariant terms $\mathbf{S}_i \cdot \mathbf{S}_j$ obtained without SOC thus undergo a simple transformation

$$\mathbf{S}_i \cdot \mathbf{S}_j \rightarrow J_{iz} J_{jz} + 3 \left( J_{ix} J_{jx} + J_{iy} J_{jy} \right). \tag{4}$$

As a result, we can proceed with analysis of the model without SOC and include the leading correction due to SOC by introducing the above substitution to the final results. This relation implies a preference for the in-plane orientation of magnetic moments in the film, which agrees with recent experimental measurements [41] and the LDA+U analysis (see Appendix A).

Evaluating $L_z$ and $L_{x,y}$ matrices in the basis of the eigenstates of atomic Hamiltonian with SOC (see Appendix C for details), we estimate the the effective $g$ factors for the lowest $J = 1$

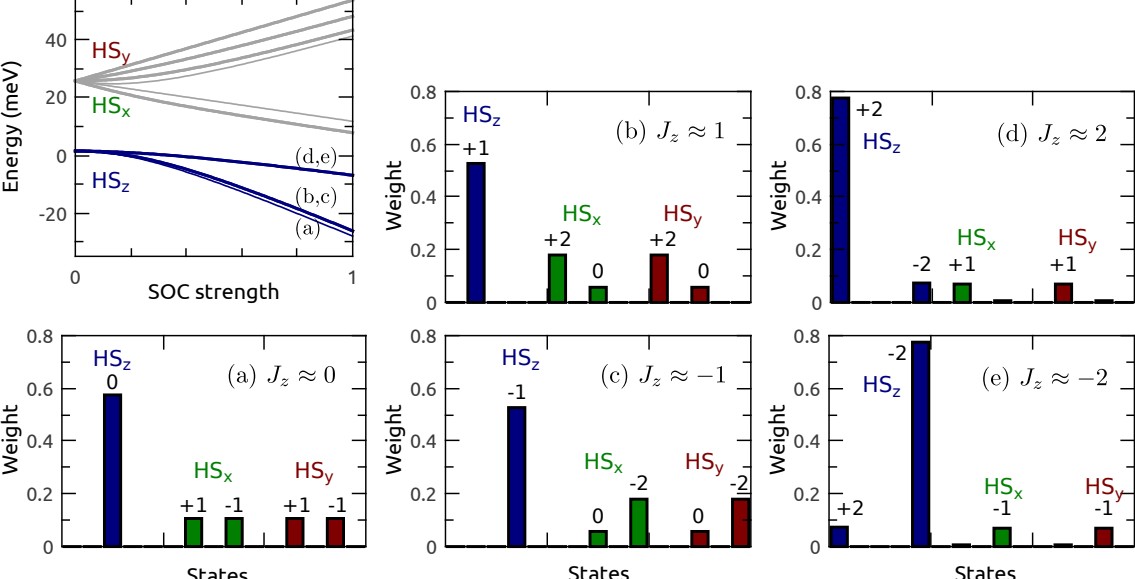

Figure 4: Splitting of atomic energies by SOC, where the maximal strength corresponds to $\zeta = 56$ meV. In this limit, the structure of 5 lowest states is shown by plotting weights of the states (a)-(e). The labels and numbers on the top of color bars correspond to the orbital character and spin projection $s_z$ in the cubic-harmonics basis ($\zeta = 0$), respectively.

triplet from the the operator relation $\mathbf{M} = \mu_{\mathrm{B}}(2\mathbf{S} + \mathbf{L}) = \mu_{\mathrm{B}}\mathbf{g}\mathbf{J}$. The obtained $g$ values are almost isotropic, $g_{zz} \approx 2.73$ and $g_{xx,yy} \approx 2.64$.

## 3.3 HS-only model

Having established the stability of HS bi-exciton with respect to HS $\longleftrightarrow$ IS-IS dissociation, we eliminate IS excitons and formulate a model $\mathcal{H}_C$ in terms of renormalized HS bi-excitons. These states can be viewed as a local HS core surrounded by IS fluctuations on nn sites. While multi-particle interactions are possible due to non-local structure of the renormalized HS states, we restrict ourselves to pairwise interactions and assume interactions involving simultaneously three or more particles to be less important. To estimate the parameters of $\mathcal{H}_C$, we diagonalalize $\mathcal{H}_B$ in the subspace with two HS$_{\hat{z}}$ bi-excitons in various geometries and analyze the spectrum and characteristics of the lowest eigenstates.

**Interaction in the $xy$ plane.** To analyze the in-plane HS-HS interactions, we choose 2D geometry, in which the two HS$_{\hat{z}}$ are confined to the $xy$ plane. Neglected IS fluctuations in the $z$-direction, which do not contribute to $xy$ interactions, amount to about 5% of the HS$_{\hat{z}}$ wavefunction. Hence, the interactions may be overestimated by 5% in 2D geometry with respect to the 3D one, a correction that is well beyond the expected accuracy of our treatment.

The confinement of IS$_{\hat{x}}$ and IS$_{\hat{y}}$ excitations to lines parallel to $y$ and $x$ axis, respectively, simplifies the calculations significantly: it allows to work in geometries corresponding to the pairs of lines parallel to the axes (with a somewhat different treatment of the situation, where the two HS states appear on the same line), see Fig. 2(e)-(g). The interaction strength for a given HS separation is obtained simply as the difference between the ground-state energy for a given two-HS geometry and twice the energy of a single HS state on a lattice of the same size.

First, we analyze the interactions in the fully FM polarized configuration, $S_z = 4$, in geom-

Table 2: Values for the HS-HS interaction amplitudes in meV units in the effective model (5) at two different amplitudes of $U/t$.

| $U/t$ | $|i-j|$ | 1 | 2 | 3 | 4 | | 1 | 2 | 3 | 4 | | nnn |
|---|---|---|---|---|---|---|---|---|---|---|---|---|
| 8 | $V_{ij}^{(x,y)}$ | 256 | 8.25 | 0.26 | 0.012 | $V_{ij}^{(z)}$ | 286 | 4.63 | 0.11 | 0.005 | $J_{\text{ex}}$ | -1.16 |
| 10 | | 277 | 5.57 | 0.11 | 0.003 | | 332 | 3.18 | 0.04 | 0.001 | | -0.66 |

etry with periodic boundary conditions and the linear size $L = 8$. The results are summarized in Table 2. The nn interaction is strongly repulsive. It is dominated by the explicit nn HS-HS interaction in $\mathcal{H}_B$, see Eq. (3) and Table 1. The AFM spin-dependent part does not overcome the spin-independent repulsion, thus the nn interaction remains strongly repulsive irrespective of spin configuration.

The second-neighbor (nnn) interaction along each of crystallographic axes is also repulsive. The origin of this repulsion is the hard-core constraint on the IS states of the same orbital character in $\mathcal{H}_B$, which restricts the virtual IS fluctuations. Since the constraint applies irrespective of spin projection, the interaction is spin-independent. The interactions at longer distances are repulsive for the same reason, but their amplitude is negligible at $U/t \geq 8$ due to localization of the IS cloud to nn sites. The interaction between HS bi-excitons that are not on the same line is negligible for the same reason, except for the nnn interaction, which requires a special treatment.

The ground state of the configuration with the two corners of a square plaquette, see Fig. 2(e), is a quasi-doublet, which originates from the HS bi-excitons exchanging the corners of the plaquette. Comparing the energies of the doublet with the energy of single HS bi-exciton as shown in Fig. 5(a), we obtain an attractive diagonal interaction as well as the amplitude of the pair-hopping $t_p$. Both the attractive nnn interaction and the pair-hopping have their origin in the on-site attraction between $IS_{\hat{x}}$ and $IS_{\hat{y}}$, which can meet in the "empty" corner of the plaquette. Since the on-site attraction is strongly FM, we expect the diagonal interaction as well as the pair-hopping to be strongly spin-dependent.

To investigate the spin structure of the diagonal nnn interactions, we perform calculations in the subspaces with total spin projections $S_z$=4, 3 and 2. The calculations are performed on a $2 \times 2$ plaquette with open boundary conditions. The results summarized in Fig. 5 reveal that: i) The spin-independent part of the nnn interaction (and pair-hopping) is small. ii) The spin exchange interaction is FM and well approximated by $\mathbf{S}_i \cdot \mathbf{S}_j$. iii) The pair hopping term has more complicated dependence on $\mathbf{S}_i \cdot \mathbf{S}_j$ and is large only for parallel spins. We point out that the mechanism of the attractive FM nnn interaction is similar to the classical 90° Goodenough-Kanamori superexchange, if one replaces the virtual excursions of electrons to the common anion site with virtual excursion of IS excitons to the common LS site.

According to the listed observations, we introduce an extended Heisenberg model in terms of the dressed HS states only (on the top of the LS vacuum),

$$\mathcal{H}_C^{xy} = \mu \sum_i n_{i,h} + \sum_{i>j} V_{ij} n_{i,h} n_{j,h} + J_{\text{ex}} \sum_{\langle\langle ij \rangle\rangle} \mathbf{S}_i \cdot \mathbf{S}_j + \frac{t_p}{4} \sum_{\langle\langle ij \rangle\rangle} \varphi(\mathbf{S}_i \cdot \mathbf{S}_j) \tau_i^+ \tau_j^+ \tau_{i'}^- \tau_{j'}^-, \quad (5)$$

where $n_{i,h} = (1 + \tau_i^z)/2$ are the HS density operators in the (HS,LS) basis, $\tau^r$ ($\tau^\pm$) are the Pauli matrices (ladder operators) representing pseudospin-1/2 operators in the same basis, $\mathbf{S}_i$ are the conventional spin $S = 2$ operators of the HS state, $\langle\langle ij \rangle\rangle$ indicates a summation over the nnn sites along two diagonals. While the indices $i$ and $j$ correspond to the sites on one diagonal in a plaquette, the sites $i'$ and $j'$ are their mirror images placed on the perpendicular one. The first term corresponds a single HS bi-exciton with the excitation energy $\mu$, the second one describes repulsion of HS bi-excitons along the crystallographic axes with characteristic

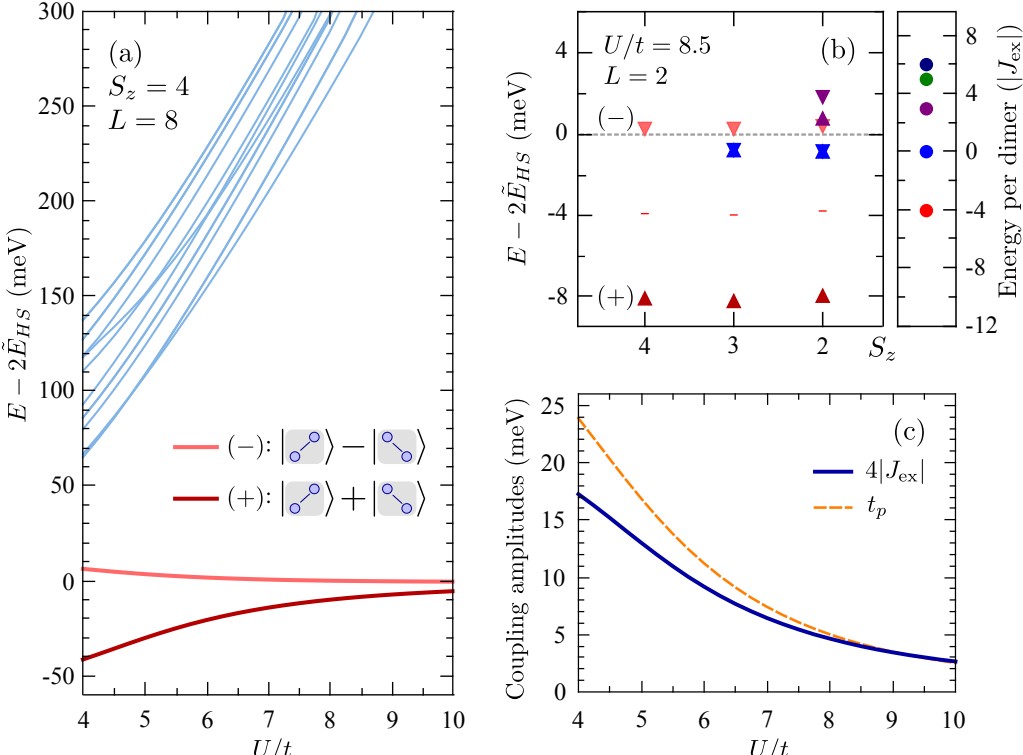

Figure 5: (a) Lowest eigenenergies for a two-HS configuration shown in Fig. 2(e) in the subspace $S_z = 4$ and $L = 8$; (b) their spin-dependent structure for the reduced system size $2 \times 2$ sites compared to the eigenenergies of the two-site dimer in the spin-2 Heisenberg model with the FM coupling $J_{ex}$. (c) Extracted values of $J_{ex}$ and $t_p$ as functions of $U/t$.

values given in Table 2. The third term is the diagonal nnn interaction and the fourth is the pair-hopping on the square plaquette.

**Interaction out of the $xy$ plane.** Extraction of the out-of-plane interaction parameters is cumbersome due to the mobility of HS bi-exciton in the $z$-direction. The repulsion along $z$-axis has similar characteristics and origin as the repulsion along the $x/y$-axis, see Table 2. There is a strong nn repulsion, originating from the bare HS-HS interaction in $\mathcal{H}_B$, and a sizable nnn repulsion, while the interaction between more distant neighbors is negligible. The mechanism of attractive diagonal interaction $J_{ex}$ present in $xy$ plane is not active on $xz$ and $yz$ plaquettes. The reason is that the HS-IS repulsion relevant in this case $J_{2x\hat{y}}^{(k)}$ is much stronger than the repulsion $J_{2x\hat{x}}^{(k)}$ relevant for the $xy$ plaquette. Simplified calculations on the $2 \times 2$ $xz$ ($yz$) plaquette suggest that the diagonal interaction is an order of magnitude weaker ($\sim 0.1$ meV). To obtain a reliable spin structure of this weak interaction, explicit inclusion of SOC may be necessary and is not attempted here.

## 3.4 HS order on the lattice

Next, we discuss ordering of the HS bi-excitons on the lattice. So far we have used the fact that the number of excitons per orbital flavor is approximately conserved, thus performed calculations for specific exciton-charge sectors without need to specify the energy of HS bi-exciton with respect to the LS vacuum, $\mu$ in Eq. (5). However, the energy $\mu$ is crucial for determination of the global ground state. Experimental estimates of $\mu$ in bulk LaCoO$_3$ yield

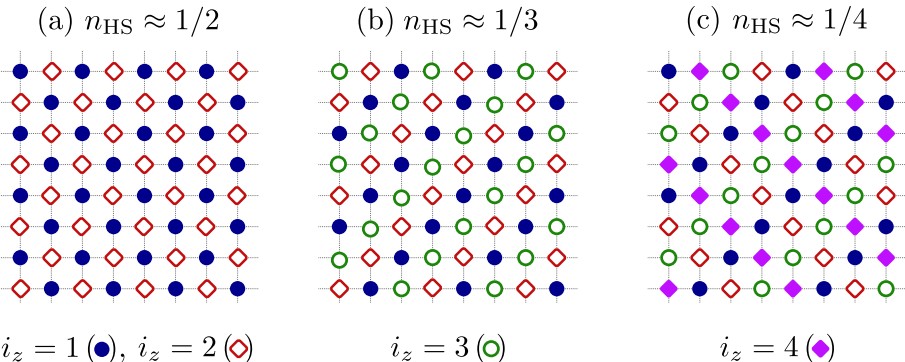

Figure 6: Configurations of HS states in the 3D lattice at fillings 1/2, 1/3, and 1/4, where the states, which occupy 2D layers with different $i_z$, are denoted by different symbols.

10–20 meV [18, 42]. To calculate $\mu$ with the desired meV accuracy from first principles is unrealistic (unlike the other coupling constants that are effectively ratios of small and large parameters).

Therefore, we base our discussion on the experimentally observed evolution of LaCoO$_3$ with increasing lateral strain from non-magnetic bulk to ferromagnetic LaCoO$_3$/SrTiO$_3$ films. The strain-induced tetragonal crystal field causes splitting of the HS multiplet into the low-energy HS$_{\hat{z}}$ singlet and the high-energy HS$_{\hat{x},\hat{y}}$ doublet, while the energy of the HS$_{\hat{z}}$ excitation above the LS state decreases, see Fig. 1. As a result, the excitation energy $\mu$ of the dressed HS$_{\hat{z}}$ bi-exciton also decreases from its bulk value.

Considering the ground state of the model (5) as a function of decreasing $\mu$, we start from an empty (LS) lattice. Upon reaching a critical $\mu_c$, the nnn interactions will lead to formation of diagonal ferromagnetic chains separated by at least two empty sites, see Fig. 6, due to the strong nn and nnn repulsive interactions. Neglecting the pair-hopping terms and replacing the Heisenberg exchange with the Ising one, the model (5) reduces to a generalization of the classical Blume-Emery-Griffiths model [43]. While the pair hopping plays a role in determining $\mu_c$ and chain separation in a single $xy$-layer problem, we conjecture that the pair hopping is strongly inhibited in the 3D problem due to strong nn repulsion along the $z$-axis. Neglecting the pair-hopping, we obtain $\mu_c = 4J_{\text{ex}}$ for a single layer problem. By reducing $\mu$ below $\mu_c$, the FM chains start to form with a separation dictated by the repulsive interactions.

To proceed with the 3D problem, we assume identical order in all layers, i.e., we only have to determine the type of stacking. Considering nn and nnn repulsion along $x$, $y$ and $z$ axes yields the 1/3 HS-filling of the lattice with the stacking shown in Fig. 6(b). Note that this order breaks the tetragonal symmetry. Another possible stacking obtained at filling 1/4 is shown in Fig. 6(c). In this case, the chains in the adjacent planes are mutually perpendicular and the tetragonal symmetry is preserved. The $(1/4, 1/4, 1/4)$ periodicity is consistent with the finding of Ref. [11]. We point out that the orders at 1/3 and 1/4 filling are not distinguished by the out-of-plane nnn interactions. The larger spacing between the chains along the $z$-axis at 1/4 filling may be a consequence of the quantum-mechanical effects that we did not consider in this subsection, i.e., the in-plane pair-hopping or hopping along the $z$-axis estimated above.

It is clear that lowering of $\mu$ favors higher lattice fillings. This can be achieved either by applying more strain or applying a strong magnetic field. This way one can proceed from 1/4 via 1/3 to 1/2 filling of the lattice. To achieve $n_{\text{HS}} = 1/2$, the magnetic energy must overcome the nnn repulsion $V_2^{(x)} + V_2^{(y)} + V_2^{(z)}$ per atom (see Table 2). With the present estimates of the $g$ factor and the nnn amplitudes, the necessary field falls to the 90–130 T range. A further increase to the regime $n_{\text{HS}} > 1/2$ requires to overcome the nn repulsion in the 200–300 meV

range, thus cannot be achieved with the available magnetic fields.

Finally, we comment on the HS-HS FM exchange reported in some DFT studies, where the local-density approximation (LDA+U) was employed. The corresponding calculations by the same approach (see Appendix A) show that there is no universal 90° FM exchange mechanism at work and that the sign of the effective coupling depends strongly on the local interaction parameters. Indeed, the setting on Co sublattice in the spin-state ordered phase is not analogous to two cations hybridizing with two orthogonal orbitals on the common anion, for which the 90° Goodenough-Kanamori rule applies [28]. In the spin-state ordered phase, the two HS ions hybridize with a common $x^2 - y^2$ orbital on the LS site and, therefore, the AFM exchange should be expected, which is the case for effective interaction below 7 eV (neglecting the structural relaxation).

## 3.5 Relationship to bulk LaCoO$_3$

An effective Hamiltonian for bulk LaCoO$_3$ can be constructed along the same lines as the present model for the strained film. Given the dominant HS–HS repulsion on nn bonds, the question arises why does the bulk system remains uniform [44] despite a high concentration of HS excitations [18, 42, 45] at temperatures that are order of magnitude lower than the nn interaction. We point out that a spin-state ordered state (checkerboard arrangement of HS and LS sites) is expected at these HS concentrations [46–48] even if the $T = 0$ ground state is a pure LS state [22, 49, 50]. To understand why the bulk LaCoO$_3$ at elevated temperature does not exhibit the checkerboard spin-state order or a more complex order as in the strained films, while it stays insulating, is one of the central questions concerning this material. We suggest that the difference between bulk LaCoO$_3$ and the strained films is the number of available HS states. While the repulsion between the HS excitations with the same orbital character is strong in all directions, the repulsion between HS excitations with different orbital character is weaker [2] and can be further reduced by anti-ferromagnetic orientation of their spin moments [26]. We hypothesize that in bulk LaCoO$_3$ the orbital degeneracy allows the HS excitations come closer to each other and prevents formation of the spin-state ordered phase. The tetragonal crystal field of the strained films selects the HS$_{\hat{z}}$ state as the only relevant low-energy excitation, which results in the spin-state ordered configuration supporting FM order. It should be pointed out that the mechanism leading the present FM state is very different from the ferromagnetism of metallic La$_{1-x}$Sr$_x$CoO$_3$.

## 4 Summary

We have studied the insulating ferromagnetism of LaCoO$_3$ films associated with the state disproportionation on Co sites. We have constructed a low-energy effective model in two steps of strong-coupling expansion. In the first step, we convert the multi-orbital Hubbard model into a model of strongly interacting mobile IS excitons and localized HS bi-excitons. In the second step, we eliminate the IS excitons to obtain a model of quasi-immobile HS bi-excitons. The approach allows us to estimate the relative strength and magnitude of various couplings in the model.

The key observations are the following. The HS state can be viewed as a bound pair of two IS excitons, which undergo virtual hopping to the neighbor sites. These fluctuations mediate HS-HS interactions beyond the nn bonds. We find that HS-HS interactions along $x$, $y$, and $z$ axes are repulsive, while the nnn $(\pm1, \pm1, 0)$ interaction is attractive for ferromagnetic orientation of the HS moments. We identify this interaction as the origin of the FM state of

---

[2]This effect is related to the strong orbital and directional dependence of the IS-IS interaction in Table 1.

strained LaCoO$_3$ and thus diagonal FM HS chains can be viewed as the basic building blocks of the FM state. The strong nn and nnn repulsion along crystallographic axes is responsible for the separation of HS chains by at least two nominal LS sites.

We point out that the HS-HS interaction mediated by IS hopping is not captured by LDA+U [13, 27, 28, 47] or LDA plus the dynamical mean-field theory [26, 48, 51, 52] calculations, while both of these approaches take into account the superexchange processes mediated by electron hopping. We have also argued that the main effect explaining the qualitatively different behavior of bulk LaCoO$_3$ and strained films is the orbital degeneracy of the HS multiplet, which is removed in the films by the tetragonal crystal field.

## Acknowledgements

The authors thank Atsushi Hariki, Jiří Chaloupka, and Dominik Huber for fruitful discussions, Juan Fernández Afonso for initiating the DFT calculations, and Anna Kauch for critical reading of the manuscript. Part of calculations was performed at the Vienna Scientific Cluster (VSC).

**Funding information.**    This work has received funding from the European Research Council (ERC) under the European Union's Horizon 2020 research and innovation programme (grant agreement No. 646807-EXMAG). K.-H.A. is also supported by National Research Foundation (NRF) of Korea Grant No. NRF-2016R1A2B4009579 and No. NRF-2019R1A2C1009588.

## A   DFT results

**Computational details.**    We performed DFT calculations based on the exchange-correlation functional of the local density approximation (LDA) with the all-electron full-potential code WIEN2K [31]. For simulating the strained lattice structure, the centrosymmetric $P4/mmm$ space group is employed with $a = 3.89$ Å and $c = 3.79$ Å being chosen for the lattice parameters. The basis size is determined by $R_{MT} \times K_{max} = 7.0$, with the muffin-tin radii of 2.5 for La, 1.91 for Co, and 1.65 for O in units of Bohr radius. To simulate the magnetic phases in the LaCoO$_3$ system, we construct a $4 \times 4 \times 4$ supercell. Fig. 7 shows three configurations: FM, AFM, and paramagnetic (PM). The AFM order is set in all **a**, **b**, and **c** directions, resulting a structure of the space group $C2/m$ (no. 12). The Brillouin zone is sampled with a regular mesh containing up to 78 irreducible $k$ points, which is sufficient to obtain a converged spin magnetic moment.

The LDA+U approach within the fully-localized limit (FLL) [53] is performed to investigate the magnetic phases of Co sites due to strong correlation effects. To this end, we employed the effective Hubbard parameter defined by $U_{eff} = U - J$ with $J = 0$, which is widely used in the FLL method. For SOC effects, four kinds of magnetization directions, [001], [100], [110], and [1$\bar{1}$0] are considered. In the AFM configuration, the direction of [110] is parallel to the stripe, while [1$\bar{1}$0] is perpendicular.

**Effects of $U$ and SOC.**    The differences of the total energies of three configurations in a wide range of $U_{eff}$ are shown in Fig. 8. We have confirmed that the solutions obtained for all $U_{eff}$ values (ranging from 4 to 8 eV) consist of HS and LS states only, while the IS contributions are negligibly small. As $U_{eff}$ increases, the magnetically-ordered configurations become more favorable than the PM one. At $U_{eff} < 7$ eV, the AFM instability becomes dominant. $U_{eff}$ values in this range are reasonable: $U_{eff} = 3.8$ eV for Ref. [29] and $U_{eff} = 5.4 - 7.0$ eV for

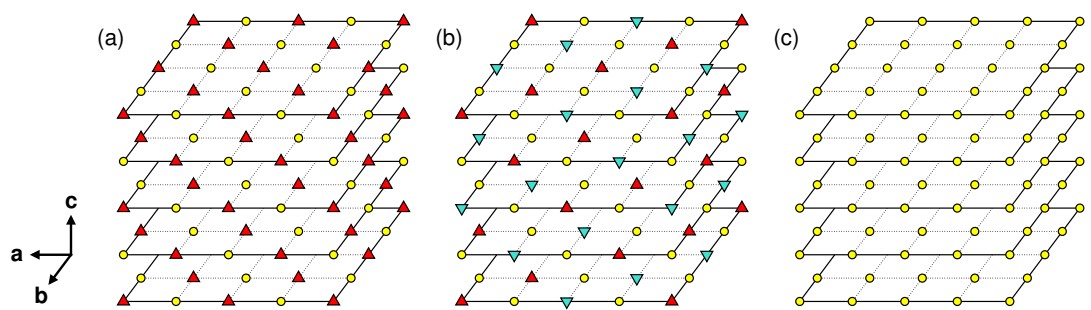

Figure 7: Magnetic configurations of (a) FM, (b) AFM, and (c) PM, which are analyzed within the DFT approach. Here, the spin states are indicated for HS↑ (red triangle-up), HS↓ (blue triangle-down), and LS (yellow circle).

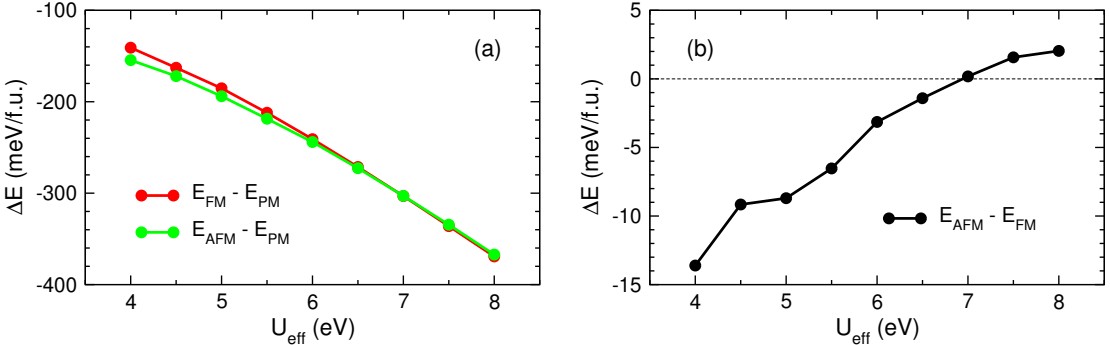

Figure 8: Energy differences as functions of $U_{eff}$ for (a) $E_{FM} - E_{PM}$ (red), $E_{AFM} - E_{PM}$ (green), and (b) $E_{AFM} - E_{FM}$.

Ref. [28] to study the HS-LS mixture. At $U_{eff} > 7$ eV, the FM configuration has the lowest energy, but the energy gain with respect to the AFM setting remains small (about 2 meV/f.u.).

Fig. 9 shows the energies at different magnetization directions in the presence of SOC. Our results indicate that the easy axis lies on the **ab** plane and can not be aligned to the **c** axis. For the AFM phase, the preffered orientation corresponds to the $[1\bar{1}0]$ direction. Compared to Fig. 8(a), SOC results in an increase of the energy difference $E_{PM} - E_{FM/AFM}$ by about 20 meV that agrees well with the calculated energy lowering of $HS_z$ states in Fig. 4.

## B    Coupling constants

The matrix elements determining the free-particle local and hopping terms in Eq. (1) are obtained from the Wannier projection on the Co $d$-only model. In the absence of SOC, the local matrix is diagonal in the cubic harmonics basis. Denoting the states $d_{xy}$, $d_{yz}$, $d_{zx}$, $d_{x^2-y^2}$, and $d_{3z^2-r^2}$ by the indices 1–5, respectively, the corresponding diagonal elements in eV units are: $\varepsilon_1 = 0$, $\varepsilon_{2,3} = 0.028$, $\varepsilon_4 = 1.591$, and $\varepsilon_5 = 1.75$. The nn hopping term, $\mathcal{H}_t = \sum_{\mathbf{i},\mathbf{e}} \sum_{\kappa\lambda} t_{\kappa\lambda}^{(\mathbf{e})} c_{\mathbf{i}\kappa}^{\dagger} c_{\mathbf{i}+\mathbf{e}\lambda}$, where the indices $\mathbf{i}$ and $\mathbf{e}$ have the same meaning as in Eq. (2), has equivalent elements in the spin-up and spin-down sectors and no mixing between them. Due to spatial symmetries, the structure of the nonzero matrix elements $t_{\kappa\lambda}^{(\mathbf{e})}$ in the orbital domain has also a compact form. In particular, for the nn hopping amplitudes in the $xy$ plane, the diagonal elements are $t_{11}^{(x)} = t_{11}^{(y)} = -0.163$, $t_{22}^{(x)} = t_{33}^{(y)} = -0.028$, $t_{33}^{(x)} = t_{22}^{(y)} = -0.141$, $t_{44}^{(x)} = t_{44}^{(y)} = -0.442$, and $t_{55}^{(x)} = t_{55}^{(y)} = -0.165$, while there are nonzero off-diagonal elements only between the $e_g$ states, $t_{45}^{(x)} = t_{54}^{(x)} = -t_{45}^{(y)} = -t_{54}^{(y)} = 0.269$. Along the $z$ axis, the nn

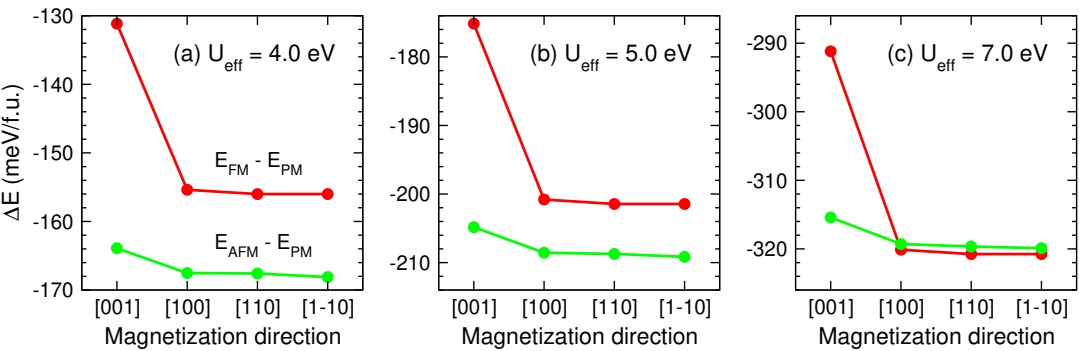

Figure 9: Energy differences at various magnetization directions of SOC for (a) $U_{eff} = 4$ eV, (b) $U_{eff} = 5$ eV, and (c) $U_{eff} = 7$ eV.

hopping amplitudes have only diagonal matrix elements, $t_{11}^{(z)} = -0.032$, $t_{22}^{(z)} = t_{33}^{(z)} = -0.198$, $t_{44}^{(z)} = 0.002$, and $t_{55}^{(z)} = -0.61$.

In the interaction term of the Hamiltonian (1), we employ the Slater parameterization of the intra-atomic electron-electron interaction matrix $U_{\kappa\lambda\mu\nu}(F_0, F_2, F_4)$ with the interaction strength $F_0 = 2.1$ eV, the Hund's coupling $\tilde{J} = (F_2 + F_4)/14 = 0.66$ eV, and the ratio of Slater integrals $F_4/F_2 = 0.625$ [54]. These interaction parameters were found to provide a good quantitative agreement with the RIXS measurements in LaCoO$_3$ [22, 23]. The employed amplitude of SOC $\zeta = 56$ meV is based on the electronic-structure analysis in LaSrCoO$_4$ [55].

## C  Spin and angular momentum matrices

In the presence of SOC with the amplitude $\zeta = 56$ meV and in the basis spanned by $J = 1$ states with $J_z = \{1, 0, -1\}$, see Figs. 4 (b), (a), and (c), respectively, the spin and angular momentum matrices take the form

$$S_z = \begin{pmatrix} 1.245 & 0 & 0 \\ 0 & 0 & 0 \\ 0 & 0 & -1.245 \end{pmatrix}, \qquad L_z = \begin{pmatrix} 0.245 & 0 & 0 \\ 0 & 0 & 0 \\ 0 & 0 & -0.245 \end{pmatrix},$$

$$S_x = \begin{pmatrix} 0 & 1.141 & 0 \\ 1.141 & 0 & 1.141 \\ 0 & 1.141 & 0 \end{pmatrix}, \quad L_x = \begin{pmatrix} 0 & -0.418 & 0 \\ -0.418 & 0 & -0.418 \\ 0 & -0.418 & 0 \end{pmatrix}.$$

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
