# Peer review of "Ferromagnetism of LaCoO$_3$ films"

_SciPost Physics, doi:SciPost Phys. 8, 082 (2020)_

## Round 1 · Referee Report · Alexander Tsirlin · 2020-5-2

Strengths
This paper suggests a possible origin of ferromagnetic behavior in the widely studied tensile-strained LaCoO3 thin films. It relies on a sophisticated computational method that involves a carefully designed and thoroughly rationalized hierarchy of approximations to arrive at the minimum model for magnetic interactions between the HS Co3+ ions.
Weaknesses
Not a weakness per se, but the paper is hard to read for someone who is not previously aware of the LaCoO3 physics
Report
Bulk and strained LaCoO3 are popular materials that have been subject to an extensive scrutiny, both experimentally and theoretically. They are probably as complex as a stoichiometric transition-metal compound can get, with an intricate interplay of spin, orbital, and even charge degrees of freedom leading to non-trivial temperature-, magnetic field-, and strain-driven changes in the electronic structure and magnetic response. In the present work, the authors focus on one intriguing aspect of this system -- ferromagnetism of tensile-strained films -- and derive a plausible microscopic scenario for ferromagnetic interactions between the high-spin Co3+ ions. This work clearly advances our understanding of the LaCoO3 physics and will be instrumental in interpreting multiple experimental results reported in recent years (at least 6 papers published in 2018-19). I support publication in SciPost Physics but would like to raise several critical comments, as I believe that several aspects of the work may need clarification, and an overall accessibility of the paper could be improved.
1. Lattice changes are known to occur in bulk LaCoO3 in response to the LS-HS transition. Some lattice relaxation should take place in thin films too. For example, [PRL 120, 197201 (2018)] reports the formation of a superstructure in LaCoO3 films on STO with exactly the same amount of tensile strain as the authors consider. Several other publications [Nano Lett. 12, 4966 (2012); Chem. Mater. 26, 2496 (2014); Phys. Rev. Materials 3, 114409 (2019)] also discuss the structural evolution of tensile-strained films, but regrettably are not even mentioned in the manuscript. I appreciate that the authors choose a different approach and analyzed a model thin film with the simplest possible structure instead of considering intricacies of the structural relaxation, but even in this case two aspects need to be clarified:
i) Is the crystal structure used in this manuscript simply a perovskite squeezed along 'c', or was a structural relaxation attempted? Are all Co-O-Co angles equal to 180 degrees? Is Co in the center of the octahedron? These details should be mentioned. Note that the PRM'2019 paper indicates the P4mm symmetry of a surface layer with the inversion symmetry lost. Do the authors consider this situation, or the centrosymmetric P4/mmm one ("simple squeezed perovskite")?
ii) Structural relaxation may also stabilize the ferromagnetic state. The small energy preference of the AFM state in DFT+U can be probably overcome by changes in the Co-O-Co angles and superexchange. I believe that this possibility should be left open, unless the authors have robust evidence against it.
Additionally, the aforementioned references certainly deserve to be cited.
2. The authors conclude that the low-energy behavior of the system maps onto an Ising-like spin Hamiltonian, but this Hamiltonian is not written explicitly. Instead, readers are confronted with Eq. (5), which is not easy to analyze, especially for non-experts. I understand that the exact form of the spin Hamiltonian will probably depend on the HS filling, but it may be still good to give examples (e.g., for the filling factors shown in Fig. 6) and make qualitative statements, especially about the easy direction. Is it in-plane or out-of-plane? Could spin direction be an experimental fingerprint of the proposed microscopic scenario?
3. Related to the previous question, [Sci. Advances 5, eaav5050 (2019)] reports magnetization of a similarly strained thin film probed for different field directions by neutron reflectometry. Can these results be connected to the Ising-like anisotropy of the spin Hamiltonian? In any case, this reference also deserves to be mentioned.
4. The presentation style is a more subjective issue, especially for SciPost Physics, which is, as of now, a predominantly theory journal. Nevertheless, in my opinion the present manuscript is one example of a SciPost Phys. publication that will be (or at least can be) very interesting to the experimental audience, but most experimentalists are likely to be lost already in the beginning where HS-Co3+ is introduced as a bi-exciton. In my opinion, the paper could become more accessible if it contained a somewhat longer introduction and a cartoon picture of the electronic states of Co, probably derived from the existing Fig. 1. In its current version, this figure contains some kind of a level scheme, but it is not labeled as such in the figure caption. I would suggest showing a similar level scheme for the thin film and bulk in order to give readers a simple picture of HS-Co3+ states being thermally-activated excitations in the bulk but condensing in the thin film. I believe it may help the readers.
5. In Sec. 3.5, the authors make a far-reaching statement that the absence of a superstructure (i.e., the uniform distribution of HS-Co3+) in the bulk is caused by the different orbital configuration. However, a significant fraction of HS-Co3+ states appears in the bulk only at elevated temperatures, whereas in the thin film they are present even at low T. Can entropy also play a role?
Requested changes
1. Add missing references
2. Explain details of the DFT calculations
3. Modify the discussion to reflect the possible influence of lattice effects
4. Add details on the Ising-like spin Hamiltonian and its easy axis
5. If possible, make the paper more accessible for a broader audience

---

## Round 2 · Referee Report · Anonymous (Referee 2) · 2020-5-14

Report

The authors made the revision in accordance with my previous report, so I am glad to recommend this manuscript for publication

---

## Round 2 · Referee Report · Anonymous (Referee 3) · 2020-5-23

Report

The authors present a theoretical study of emerging long-range ferromagnetic ordering in tensile strained LaCoO3 using numerical simulation. Building a model accompanied with a series of approximation to account electron correlation effects originating from Co3+ ions, authors tried to answer ferromagnetism and atomic-scale inhomogeneity in LaCoO3 films. A qualitative agreement with experimental evidence is claimed.

In my opinion the results are interesting and correct. The authors present manuscript in systematic way, but yet not well in pedagogical manner. The paper can be published as it is. My suggestion for the authors is just to double-check the paper for possible typos, and also give a clear reference about “Site 0 is the crossing point” statement mentioned in the caption of figure.3

---

## Round 2 · Author Response

We thank the referee for the detailed feedback and helpful suggestions. These have been very useful for improving the manuscript. Please find below our replies in the same order as in the referee's report with a list of changes and the revised manuscript.

  1. We have added the missing references to the studies on the structural relaxation in the strained compound. One paragraph with a corresponding discussion was added in the end of the first paragraph of Sec. 2 including Refs. [35-38]. The point ii) was addressed there as well, while in response to the point i), the second sentence in the first paragraph in Appendix A was extended to specify the structural details in the DFT approach.

  2. We agree with the point raised by the referee and thus simplified Eq.(5) by writing it in terms of density operators and omitting redundant multipliers in the spin-exchange term.

  3. We have added Ref. [41] and provided an additional explanation on the in-plane orientation of magnetic moments in the second sentence below Eq.(4).

  4. To make the paper more accessible for a broader audience, we have extended the introduction part, in particular, the second and the fourth paragraphs in the corresponding section. In Figure 1, we have also provided a schematic comparison of the low-energy states in the bulk and the strained compound. To emphasize the difference to the ferromagnetism of metallic La_{1-x}Sr_xCoO_3, we also added one sentence in the end of Sec. 3.

  5. Our statement on the absence of a superstructure in the bulk compound in Sec. 3.5 refers explicitly to the origin of the low population of HS states in the T=0 limit compared to the strained compound. Therefore, in our opinion, there must be still energy-related arguments, as it is given in the text.

---

## Round 2 · List of Changes

1. Introduction was extended, in particular the text was added to the second and the fourth paragraphs of Sec. 1.
  2. Fig. 1 was updated: the right panel now contains a comparison of local energies of the lowest states in the bulk and the film structures. The text in the figure caption was updated correspondingly.
  3. One sentence with a discussion of the structural relaxation with Refs. [35-38] was added in the end of the first paragraph of Sec. 2.
  4. One sentence with a discussion of the direction of magnetic moments in the film compound with Ref. [41] was added below Eq.(4).
  5. Notations in Eq.(5) were simplified for a better accessibility.
  6. One sentence emphasizing the difference to the ferromagnetism of metallic La$_{1-x}$Sr$_x$CoO$_3$ was added in the end of Sec. 3.
  7. Details on the employed structure in the DFT approach were added in the second sentence of Appendix A.

---

## Editorial Decision

published